# Comparison of Outcomes between Minimally Invasive Lateral Approach Vertebral Reconstruction Using a Rectangular Footplate Cage and Conventional Procedure Using a Cylindrical Footplate Cage for Osteoporotic Vertebral Fracture

**DOI:** 10.3390/jcm10235664

**Published:** 2021-11-30

**Authors:** Naoki Segi, Hiroaki Nakashima, Tokumi Kanemura, Kotaro Satake, Kenyu Ito, Mikito Tsushima, Satoshi Tanaka, Kei Ando, Masaaki Machino, Sadayuki Ito, Hidetoshi Yamaguchi, Hiroyuki Koshimizu, Hiroyuki Tomita, Jun Ouchida, Yoshinori Morita, Shiro Imagama

**Affiliations:** 1Department of Orthopedic Surgery, Nagoya University Graduate School of Medicine, Nagoya 466-8560, Japan; naoki.s.n@gmail.com (N.S.); andokei@med.nagoya-u.ac.jp (K.A.); masaaki_machino_5445_2@yahoo.co.jp (M.M.); sadaito@med.nagoya-u.ac.jp (S.I.); hide.yama.627@hotmail.co.jp (H.Y.); love_derika@yahoo.co.jp (H.K.); hiro_tomi_1031@yahoo.co.jp (H.T.); orthochida@gmail.com (J.O.); bai.an9610@gmail.com (Y.M.); imagama@med.nagoya-u.ac.jp (S.I.); 2Department of Orthopedic Surgery, Konan Kosei Hospital, 137 Takayamachi Omatsubara, Konan 483-8704, Japan; spinesho@vmail.plala.or.jp (T.K.); k-satake@konan.jaaikosei.or.jp (K.S.); kenyubankara@yahoo.co.jp (K.I.); meikeihan@hotmail.com (M.T.); satoshi8005@yahoo.co.jp (S.T.)

**Keywords:** osteoporotic vertebral fracture, osteoporotic vertebral collapse, minimally invasive surgery, lateral access surgery, minimally invasive lateral corpectomy, anterior spinal reconstruction, anterior and posterior combined surgery, rectangular footplate

## Abstract

The aim of the current study was to compare outcomes between lateral access vertebral reconstruction (LAVR) using a rectangular footplate cage and the conventional procedure using a cylindrical footplate cage in patients with osteoporotic vertebral fracture (OVF). We included 46 patients who underwent anterior–posterior combined surgery for OVF: 24 patients underwent LAVR (Group L) and 22 underwent the conventional procedure (Group C). Preoperative, postoperative, and 1- and 2-year follow-up X-ray images were used to measure local lordotic angle, correction loss, and cage subsidence (>2 mm in vertebral endplate depression). In anterior surgery, the operation time was significantly shorter (183 vs. 248 min, *p* < 0.001) and the blood loss was significantly less (148 vs. 406 mL, *p* = 0.01) in Group L than in Group C. In Group C, two patients had anterior instrumentation failure. Correction loss was significantly smaller in Group L than in Group C (1.9° vs. 4.9° at 1 year, *p* = 0.02; 2.5° vs. 6.5° at 2 years, *p* = 0.04, respectively). Cage subsidence was significantly less in Group L than in Group C (29% vs. 80%, *p* < 0.001). LAVR using a rectangular footplate cage is an effective treatment for OVF to minimize surgical invasiveness and postoperative correction loss.

## 1. Introduction

The incidence of osteoporotic vertebral fracture (OVF) increases with age [1]. Some patients have difficulty returning to activities of daily living because of severe back pain or neurological complications with pseudarthrosis [2,3] even with conservative treatment, which results in OVF posing a significant cost to society. Minimally invasive treatment methods such as balloon kyphoplasty have been developed to overcome OVF. However, contraindications to balloon kyphoplasty include pedicle fracture and fracture of the posterior wall of the vertebral body diagnosed on computed tomography (CT); these cases require reconstructive surgery [4].

Although various reconstructive surgical techniques have been reported for OVF, including anterior reconstruction surgery, posterior fusion with or without vertebroplasty, and anterior–posterior combined surgery (AP surgery) [5,6,7,8], the best method has not yet been identified [9]. Anterior surgery alone results in the need for additional surgery rather than maintaining alignment and provides inadequate fixation, especially for patients with severe osteoporosis [7]. Posterior fixation is widely used, but its ability to maintain alignment remains inadequate [10,11]. To avoid correction loss and instrumentation failure, AP surgery is more effective than posterior fusion with vertebroplasty [11]. However, there are two problems with AP surgery. First, there is an increase in perioperative complications relative to those associated with other techniques [12]. Second, the risk of correction loss and nonunion still cannot be eliminated [11].

To overcome these problems, we began performing lateral access vertebral reconstruction (LAVR) combined with posterior surgery [13]. This method provides the following two useful features for thoracolumbar anterior surgeries: the LAVR approach and the cage footplate. The LAVR approach is a minimally invasive surgery (MIS) approach that enables us to reach the thoracolumbar vertebrae through a small incision using a dedicated retractor, which can reduce surgical invasiveness [14]. Moreover, instead of a conventional cage with a cylindrical footplate, a large rectangular footplate cage that provides coverage as wide as the vertebral endplate diameter, which may improve the mechanical supportability of the cage [15,16] and reduce correction losses, can be used in this surgical approach, even in patients with osteoporosis. Some surgical results have been reported [17]; however, due to the lack of comparison with the conventional anterior approach using a cylindrical cage, the effectiveness of LAVR for OVF continues to be debated.

Therefore, the aim of the current study was to compare surgical invasiveness and radiologic outcomes at 2 years after surgery for OVF between LAVR and the conventional procedure. We hypothesized that LAVR using a rectangular footplate expandable cage would provide less surgical invasiveness and better radiologic outcomes than the conventional procedure using a cylindrical expandable cage.

## 2. Materials and Methods

### 2.1. Patient Population

This retrospective study assessed 96 consecutive patients who underwent AP surgery for OVFs in patients aged 65 years or older at our institution between 2010 and 2019. Vertebral fractures caused by a small magnitude of force, such as falls from a standing position, were diagnosed as OVFs. Patients who developed neurological symptoms, such as paralysis of the lower limbs, patients with severe low back pain, and patients with symptomatic pseudoarthrosis, were eligible for surgery. Based on the classification by the German Orthopedic Association [18], the fracture type of the included patients was ≥OF 3. Exclusion criteria were high-energy trauma, pathologic fracture due to neoplasm, and rigid kyphosis. Patients for whom surgical treatment was performed for multiple fractures at the same time were also excluded. Patients were eligible if they had at least 2-year follow-up. The final analysis included 46 patients (Figure 1). None of the patients had time to start or change osteoporosis treatment before the surgery because of the severity of their symptoms. Two procedures were performed according to the periods. Conventional expandable cylindrical footplate cages (T2 Altitude, Medtronic Sofamor Danek, Memphis, TN, USA; SynCage-EX, Synthes, Paoli, PA, USA) were used in all 22 patients until January 2015 (Group C), and the expandable rectangular footplate cage (X-Core2, NuVasive, San Diego, CA, USA) was used in all 24 patients since February 2015 with LAVR (Group L) as shown in Figure 2.

### 2.2. Procedure for AP Surgery

Our AP surgery was planned as a two-stage surgery: posterior surgery was performed first, and then anterior surgery was performed 1 week later. All surgeries were performed under general anesthesia. The first posterior surgery used a pedicle screw-rod system to correct kyphotic deformity. Augmentations by hooks or sublaminar wirings using ultrahigh molecular weight polyethylene tapes were performed according to the surgeon’s preference. Autologous local bone and/or substitute bone were used for the posterior bone graft.

The anterior surgery was performed after an interval of 1 week. Our anterior approach for thoracolumbar junctional lesion has been reported previously in detail [19]. Briefly, the surgery in Group C was performed through a skin incision of up to 20 cm along the ribs to be resected around the fractured vertebrae, with diaphragmatic detachment if necessary. In contrast, the skin incision in Group L was approximately 5 cm along the ribs to be resected. Although this procedure can be performed without partial-rib resection, retractor placement between the ribs limits the expansion. Partial-rib resection enables free and safe surgical manipulation; hence, it was routinely performed. The diaphragm was minimally split to avoid injuring the pleura, and a dedicated retractor (MaXcess retractor, NuVasive, San Diego, CA, USA) was placed [14,19,20,21]. If required, the diaphragmatic attachments were partially dissected. A standard retroperitoneal approach was used for lumber lesions, and an extrapleural approach or thoracotomy was used for thoracic lesions, with Group L using the dedicated retractor. After the fractured vertebrae were exposed, the segmental arteries were ligated, the adjacent disks were resected in a standard procedure, and then the fractured vertebrae were resected to allow the cage to be installed. A cage was inserted and expanded to the desired height under fluoroscopic control. The cage expansion was retained until gentle contact with the endplates was made. Autologous iliac and rib bones and/or allograft bones were used for grafting inside and outside of the cages.

### 2.3. Postoperative Therapy

After the AP surgery, the patient wore a hard brace for a minimum of 3 months and underwent gait rehabilitation. After hospital discharge, the patients attended outpatient visits regularly and were followed up with CT in addition to radiography at 1- and 2-year follow-up.

### 2.4. Clinical and Radiologic Assessment

Standing X-ray images taken preoperatively, postoperatively, and at 1 and 2 years postoperatively were assessed to measure the local lordotic angle (LLA) and correction loss (Figure 3). The LLA was defined as the angle between the cranial endplate of the fractured vertebra and the caudal endplate of the caudal vertebra and was set to a negative value for kyphosis. Cage subsidence was defined as the endplate being recessed by 2 mm or more in the lateral X-ray image. The 1- and 2-year postoperative CT images were evaluated for bone fusion and pedicle screw loosening. Bony fusion was defined by the presence of continuous bone trabeculae or bridging bone between vertebrae in the sagittal or coronal images of the CT reconstruction images. Pedicle screw loosening was defined as the presence of a radiolucent zone around the screw.

### 2.5. Statistical Analysis

Data are presented as mean ± standard deviation for continuous variables and as number and percentage for categorical data. Statistical analyses were performed using R version 4.0.0 (http://www.R-project.org (accessed on 6 June 2021)) for the Welch two-sample *t* test, Fisher’s exact test, and Pearson’s chi-squared test. A value of *p* < 0.05 was considered significant.

## 3. Results

Of the 46 patients, the mean age was 74.1 years for Group C and 76.2 years for Group L. Bone mineral density measured at the proximal femur was −1.82 for Group C and −1.76 for Group L, and osteoporosis treatment was not performed in 86% of Group C and 96% of Group L. No statistically significant differences in demographics were noted outside of the follow-up period (Table 1). No significant difference was observed in the range of posterior fusion. A significant difference was found in the use of hooks and decompression between groups: Group L had more hooks (83% vs. 45%) and fewer decompressions (17% vs. 55%) (Table 2).

In anterior surgery, the operation time was significantly shorter (248 vs. 183 min, *p* < 0.001) and the blood loss was significantly less (406 vs. 148 mL, *p* = 0.01) in Group L than in Group C (Table 3). All five patients treated with the thoracotomy approach in Group C had a chest drain inserted. Two patients with pleural injuries during the extrapleural approach also needed a chest drain. In contrast, none of the Group L patients underwent thoracotomy, but three patients treated with the extrapleural approach and one patient treated with the transdiaphragm approach needed a chest drain due to pleural injury. With respect to postoperative respiratory complications, atelectasis was found in one patient in Group L; however, oxygen was not necessary, and the patient was only observed. In contrast, there were two patients with pneumonia and one patient with pleural hematoma requiring oxygen for several days in Group C. Furthermore, two patients in Group C suffered from postoperative pneumothorax, one of whom required chest drainage for treatment. In each group, thigh symptoms occurred in two patients and were transient. In Group C, two patients experienced anterior instrumentation failure (cage deviation), and one patient underwent revision surgery 5 months after the primary surgery. The other patient was in poor general condition and was only under observation. No anterior mechanical failure occurred for any patients in Group L.

A summary of the radiologic studies excluding the two patients with implant failure in Group C is shown in Table 4. In both groups, the postoperative LLA was corrected with significant difference from the preoperative angle (Group C, *p* < 0.001; Group L, *p* < 0.001) (Figure 4). The correction losses of LLA were 4.9° in Group C and 1.9° in Group L at 1 year postoperatively (*p* = 0.02) and 6.5° in Group C and 2.5° in Group L at 2 years postoperatively (*p* = 0.04). The number of patients who experienced correction loss of 10° or more was three in Group C and one in Group L at 1 year postoperatively (*p* = 0.32) and five in Group C and one in Group L at 2 years postoperatively (*p* = 0.08). Cage subsidence occurred in 80% in Group C and 29% in Group L (*p* < 0.001). Bony fusion was achieved in 65% of Group C and 92% of Group L at 2 years postoperatively (*p* = 0.06). Continuity of the trabecular through inside cage was confirmed in 60% of Group C and 29% of Group L (*p* = 0.04), and paravertebral bridging bone (Figure 5) was identified in 30% of Group C and 62% of Group L (*p* = 0.03).

After 2 years, the neurological status of the patients had recovered in terms of ASIA classification in the majority of patients, with 41% of Group C and 42% of Group L patients achieving unassisted ambulation (Table 5). There was no significant difference in walking ability.

## 4. Discussion

We believe that this study is the first to directly compare the effectiveness of LAVR using rectangular footplate cage with the conventional procedure using cylindrical cage for OVF. Our results showed a decrease in operation time and blood loss, as well as a decrease in complications. We also found that correction loss and cage subsidence were significantly smaller in LAVR at 1 year and 2 years postoperatively. Furthermore, the bony fusion rate was higher in LAVR. Therefore, MIS LAVR was more effective for OVF treatment.

Because of osteoporosis, strong stability is essential for the surgical treatment for OVF. Ulmar et al. [22] biomechanically demonstrated that the combined anteroposterior instrumentation is the strongest internal stabilization in all motion planes for rotationally unstable vertebral body fractures. Nakashima et al. [11] compared AP surgery and posterior fixation with vertebroplasty for OVF and demonstrated the following two aspects: (1) AP surgery is a stable spinal fixation and reduces implant failure and (2) perioperative complication rates were almost similar with both procedures. However, even in the conventional AP surgery, correction loss occurred postoperatively [11]. We considered that LAVR using a larger rectangular footplate cage, which was recently reported as minimally invasive anterior surgery [21,23,24], may overcome this shortcoming of conventional AP surgery for OVF.

In the current study, correction loss was significantly smaller and the incidence of cage subsidence was significantly lower in Group L than in Group C. Furthermore, there were two patients with instrumentation failure who should have been considered for revision surgery in Group C, but there were none in Group L. A rectangular footplate is mechanically stable compared with a cylindrical footplate [16]. The rectangular footplate cage has advantages over the conventional cylindrical cage in at least two aspects. One advantage is the mechanical supportability by the footplate that has a width as wide as the diameter of the vertebral body. A cadaveric study conducted by Hasegawa et al. [25] showed that a titanium mesh cage with a large diameter produces a significant increase in interface strength between the cage and vertebra. Moreover, Lowe et al. [26] demonstrated that the posterolateral region of the endplate provides the greatest resistance to subsidence in a human cadaveric specimen. Kreinest et al. [27] defined the safe zone on the endplate based on previous reports, showed clinical results, and conducted biomechanical studies on osteoporotic bone. Kreinest’s safe zone is also mainly located in the posterolateral area of the vertebral body next to the pedicles, and the study results showed that both the contact area between the cage and endplate or cage positioning within the safe zone did not significantly improve stability. However, the cage they used did not have a footplate as wide as the vertebral body; therefore, the current study provides novel useful results. Based on these studies, the mechanical supportability of the rectangular footplate cage results from not only the increased contact area, but also the fit of the footplate to the vertebral endplate edge, which has higher bone strength.

In addition, the rate of bony fusion was also higher in Group L. It is noteworthy that a difference was found not only in the fusion rate but also in the fusion morphology. In Group C, bone continuity was observed mainly inside the cage. This finding was because the cylindrical cage subsides in many patients, and bony fusion occurred where the cage entered the vertebral body. In contrast, in Group L, slight cage subsidence resulted in a different bony fusion form than in Group C. Vertebrae in Group L were fused not only on the approached side but also on the contralateral vertebral side with bridging bone. This finding may reflect the LAVR technique in which the lateral approach also allows dissection of the contralateral side when the disc is treated.

Correction loss is associated with kyphosis/pseudoarthrosis and increased implant failure/revision surgery; in addition, chronic pain, poor chest function, decreased appetite (gastroesophageal reflux disease [28,29]), and fatigue are more common in these patients [30]. To eliminate these potential risks, reconstructive AP surgery must prevent correction loss and achieve solid bony fusion. Hence, LAVR is better for maintaining correction and preventing these systemic problems, which may be associated with better clinical outcomes.

Another feature of LAVR is the application of the MIS lateral approach that can reduce surgical invasiveness. With a dedicated retractor, dissection of the abdominal muscle is minimized for lumbar lesions [21], and the transdiaphragmatic approach is feasible as it avoids diaphragmatic detachment in lesions of the thoracolumbar junction [19]. In this study, it is worth noting that not only were there fewer respiratory complications in Group L (4.2% vs. 23%, *p* = 0.09), but their severity was also different. Moreover, the operation time of anterior surgery was shortened. Although there is a learning curve for the MIS lateral approach, the procedure can be performed quickly with good training and experience. Based on these facts, we conclude that LAVR with a dedicated retractor can reduce surgical invasiveness.

Even though LAVR reduced the operation time, the reduction was insufficient from the perspective of minimally invasive surgery. Kreinest et al. [31] defined an operation time of >120 min as prolonged surgery in their report on anterior approach vertebral body replacement. However, in our institution, young surgeons (fellows) often operated under the supervision of an attending surgeon; therefore, not all surgeries were performed by senior surgeons, as reported by others. Furthermore, the LAVR is time-consuming in some situations. To install the rectangular footplate cage, both the cranial and caudal discs must be almost completely removed through the contralateral side; otherwise, the cage will not be able to be guided to its proper position. This task is even more delicate in osteoporotic patients because endplate damage can easily occur. Because of this technique, using the rectangular footplate cage is more time-consuming compared to using the cylindrical cage. However, in addition to the small skin incision, muscles will be easier to repair following LAVR due to their minimal disruption, including the diaphragm; this will save time overall.

However, it should be noted that in addition to the advantages of this technique, there are also some important considerations. In LAVR with a dedicated retractor, there are restrictions on access paths and surgical fields, which can lead to technical difficulties or surgical complications. With adequate training, LAVR can be performed with a 4 cm skin incision [21], but serious complications, such as nerve and organ damage, can occur without sufficient surgical skill. To avoid this potential problem, detailed anatomical knowledge is more important in LAVR than in conventional anterior surgery [32,33]. Smith et al. [14] warned that an advanced understanding of the relevant anatomy and the minimally invasive exposure in a lateral approach are essential. Certainly, LAVR reduces the operation time, blood loss, respiratory complications, and correction loss relative to those in conventional open corpectomy. However, minimal-incision and less-invasive lateral surgery may be a trade-off with the limited visualization of the surgical space and smaller working space. Correct anatomical recognition is essential to achieve success with LAVR, and the surgeon should not hesitate to achieve sufficient surgical vision and workspace if needed even in LAVR. Table 6 presents comparative results of the two procedures.

In the current study, 21 patients were excluded because of a short follow-up period, which may have caused selection bias. Six patients were in Group C (mean age of 74.5 years, one male), and 15 patients were in Group L (mean age of 77.8 years, seven males). All patients were untraceable after being transferred to other medical facilities for postoperative care and rehabilitation. Our institution is located in a rural area, and postoperative patients are often followed up at local hospitals. Patients from distant areas, especially elderly patients with OVFs, have difficulty visiting our hospital for postoperative follow-up. Missing data due to follow-up dropout is a limitation of retrospective studies; hence, prospective registry studies should be conducted to address this problem. This study had several other limitations: the number of patients was small, the follow-up period was relatively short, and the posterior surgical procedures were not completely consistent between the two groups and among the patients. It is necessary to further compare and investigate the clinical results for LAVR and the conventional approach and to evaluate the usefulness and problems of this implant.

## 5. Conclusions

Surgical invasiveness and outcomes of AP surgery for the elderly patients with OVF were compared between MIS LAVR using a rectangular footplate cage and the conventional procedure using a cylindrical footplate cage. The MIS LAVR method resulted in shorter operative time and less blood loss. Radiologically, it provided a reduction in correction loss and cage subsidence and improved the bony fusion rate 2 years after surgery. This new procedure using a cage with a novel footplate is effective for minimally invasive surgical treatment of OVF to achieve stable radiologic outcomes.

## Figures and Tables

**Figure 1 jcm-10-05664-f001:**
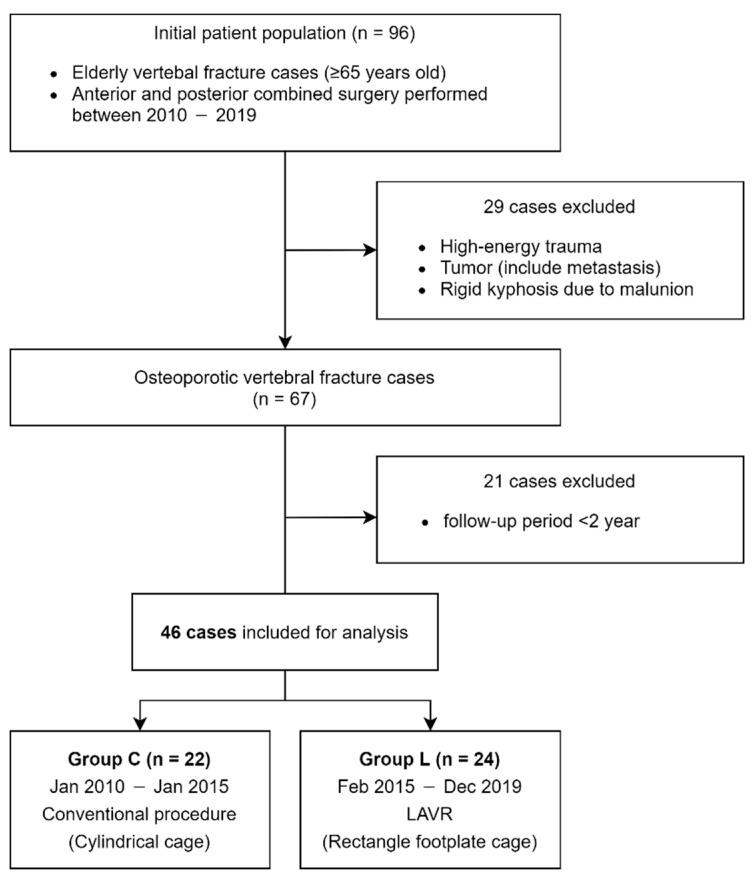
Flowchart for patient selection.

**Figure 2 jcm-10-05664-f002:**
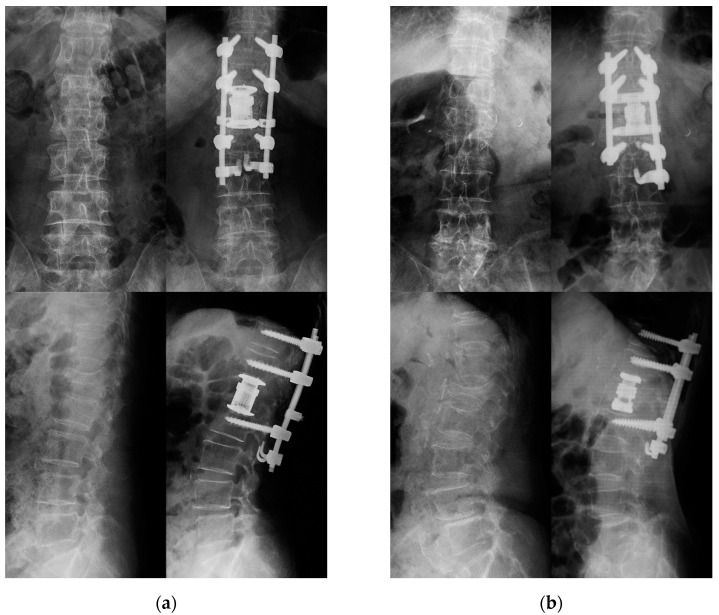
Representative pre- and postoperative radiographs of patients in (**a**) Group C and (**b**) Group L.

**Figure 3 jcm-10-05664-f003:**
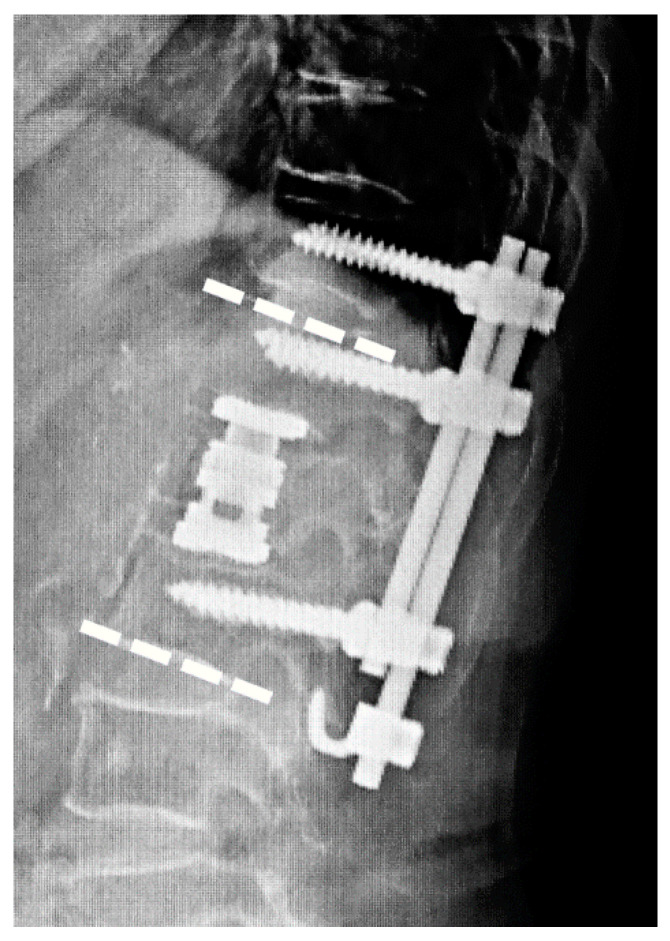
Measurements of the radiographic image. The upper and lower dashed lines indicate the local lordotic angle (LLA).

**Figure 4 jcm-10-05664-f004:**
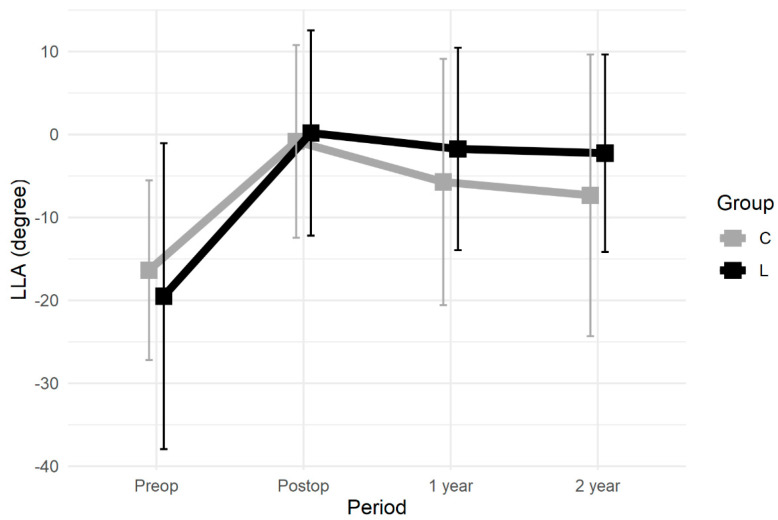
Time course of local lordotic angle.

**Figure 5 jcm-10-05664-f005:**
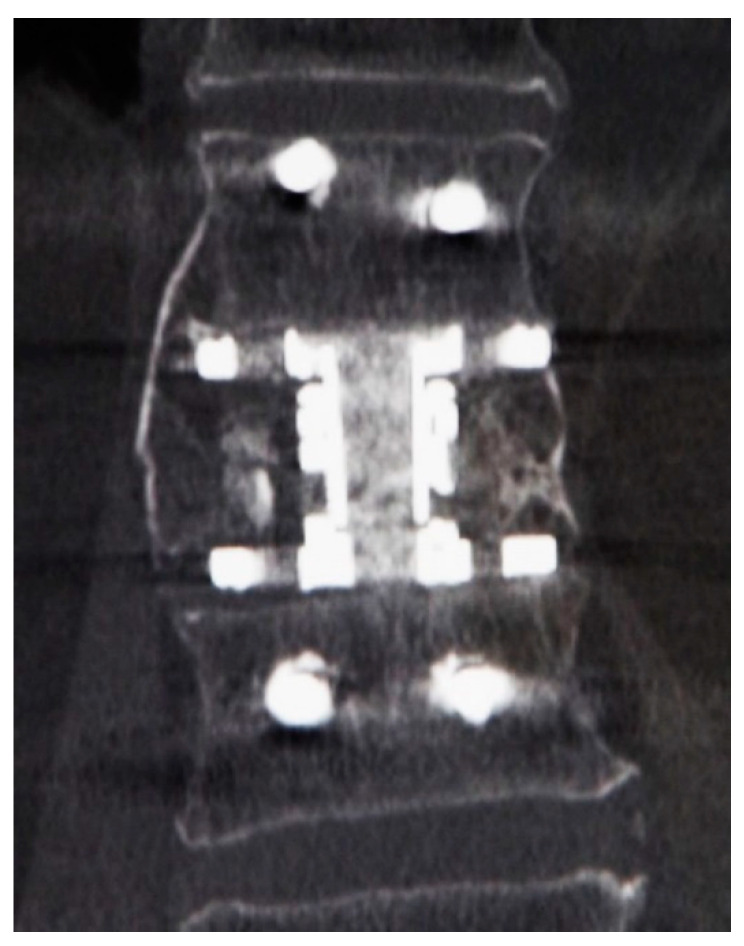
Paravertebral bridging bone formed on the opposite side of the approach (lateral approach vertebral body reconstruction (LAVR) patient).

**Table 1 jcm-10-05664-t001:** Demographics.

	Group C, *n* = 22	Group L, *n* = 24	*p* Value
Age (years), (SD)	74.1 (5.7)	76.2 (6.2)	0.24
Sex, Male	11 (50.0%)	9 (37.5%)	0.39
Postinjury period			0.67
<1 month	2 (9.1%)	4 (17%)	
more	20 (91%)	20 (83%)	
Smoking			0.11
Current	4 (18%)	2 (8.3%)	
Ex-smoker	8 (36%)	4 (17%)	
No	10 (45%)	18 (75%)	
Comorbidities			
Cardiovascular	3 (13.6%)	6 (25.0%)	0.46
Respiratory	2 (9.1%)	1 (4.2%)	0.60
Diabetes mellitus	7 (31.8%)	6 (25.0%)	0.61
Hypertension	15 (68.2%)	17 (70.8%)	0.85
Hyperlipidemia	3 (13.6%)	3 (12.5%)	>0.99
Stroke	2 (9.1%)	0 (0.0%)	0.22
Hepatic	2 (9.1%)	1 (4.2%)	0.60
Rheumatoid arthritis	0 (0.0%)	1 (4.2%)	>0.99
T-score ^1^ (SD)	−1.82 (0.77)	−1.76 (1.20)	0.85
Osteoporosis treatment			0.16
No	19 (86%)	23 (96%)	
Teriparatide	2 (9.1%)	0 (0%)	
Oral bisphosphonate	1 (4.5%)	0 (0%)	
SERM	0 (0%)	1 (4.2%)	
Lesion level			0.10
T9	1 (4.5%)	0 (0%)	
T10	0 (0%)	1 (4.2%)	
T11	0 (0%)	1 (4.2%)	
T12	5 (23%)	8 (33%)	
L1	9 (41%)	8 (33%)	
L2	5 (23%)	1 (4.2%)	
L3	0 (0%)	4 (17%)	
L4	2 (9.1%)	1 (4.2%)	
ASIA classification			0.33
C	5 (23%)	9 (38%)	
D	5 (23%)	7 (29%)	
E	12 (55%)	8 (33%)	

SERM, selective estrogen receptor modulator; ASIA, American Spinal Cord Injury Association. ^1^: T-score detected by dual-energy X-ray absorptiometry of proximal femur.

**Table 2 jcm-10-05664-t002:** Posterior surgery summary.

	Group C, *n* = 22	Group L, *n* = 24	*p* Value
Fusion range			0.44
1 above–1 below	4 (18%)	2 (8.3%)	
2 above–1 below	5 (23%)	9 (38%)	
2 above–2 below	10 (45%)	12 (50%)	
more	3 (14%)	1 (4.2%)	
Augmentation			
Sublaminar wire	15 (68%)	17 (71%)	0.85
Hook	10 (45%)	20 (83%)	0.01
Decompression	12 (55%)	4 (17%)	0.01

**Table 3 jcm-10-05664-t003:** Anterior surgery and perioperative summary.

	Group C, *n* = 22	Group L, *n* = 24	*p* Value
Anterior surgery			
Op time (min), (SD)	248 (68)	183 (41)	<0.001
Blood loss (mL), (SD)	406 (432)	148 (137)	0.01
Anterior approach			<0.001
Thoracotomy	5 (23%)	0 (0%)	
Extrapleural	11 (50%)	7 (29%)	
Transdiaphragm	0 (0%)	12 (50%)	
Retroperitoneal	6 (27%)	5 (21%)	
Chest drain ^1^	7 (44%)	4 (21%)	0.49
Complications			
Respiratory	5 (23%)	1 (4.2%)	0.09
Stroke	0 (0%)	1 (4.2%)	>0.99
Delirium	4 (18%)	3 (12%)	0.69
Thigh symptom	2 (9.1%)	2 (8.3%)	0.90
SSI (superficial)	1 (4.5%)	0 (0%)	0.48
Implant failure	2 (9.1%)	0 (0%)	0.22
Vertebral fracture	6 (27%)	5 (21%)	0.61
Osteoporosis treatment			0.47
No	11 (50%)	8 (33%)	
Teriparatide	9 (41%)	13 (54%)	
Oral bisphosphonate	1 (4.5%)	1 (4.2%)	
Denosumab	0 (0%)	2 (8.3%)	
SERM	1 (4.5%)	0 (0%)	

SSI, surgical site infection; SERM, selective estrogen receptor modulator. ^1^: Indicates the percentage excluding cases treated with the retroperitoneal approach.

**Table 4 jcm-10-05664-t004:** Summary of radiological survey.

	Group C, *n* = 20 ^1^	Group L, *n* = 24	*p* Value
LLA (°), (SD)			
Preoperative	−16.4 (10.8)	−19.5 (18.4)	0.49
Postoperative	−0.8 (11.6)	0.2 (12.4)	0.78
1 year	−5.7 (14.8)	−1.7 (12.2)	0.34
2 year	−7.4 (16.9)	−2.3 (11.9)	0.27
LLA loss (°), (SD)			
1 year	4.9 (5.0)	1.9 (2.5)	0.02
2 years	6.5 (7.6)	2.5 (3.0)	0.04
Loss over 10°, 1 year	3 (15%)	1 (4.2%)	0.32
Loss over 10°, 2 years	5 (25%)	1 (4.2%)	0.08
Cage subsidence	16 (80%)	7 (29%)	<0.001
Bony union			
1 year	9 (45%)	14 (58%)	0.38
2 years	13 (65%)	22 (92%)	0.06
Trabecula through cage	12 (60%)	7 (29%)	0.04
Paravertebral bridging bone	6 (30%)	15 (62%)	0.03
Approach side	6 (30%)	10 (42%)	0.42
Opposite side	0 (0%)	8 (33%)	0.01
PS loosening	4 (20%)	3 (12%)	0.68

^1^: Two patients who experienced implant failure (cage deviation) were excluded.

**Table 5 jcm-10-05664-t005:** Clinical status after 2 years.

	Group C, *n* = 22	Group L, *n* = 24	*p* Value
ASIA classification			0.25
D	2 (9.1%)	6 (25%)	
E	20 (91%)	18 (75%)	
Ambulation			0.96
Gait alone	9 (41%)	10 (42%)	
Gait with cane	9 (41%)	11 (46%)	
Gait with cart	2 (9.1%)	2 (8.3%)	
Wheelchair	2 (9.1%)	1 (4.2%)	

ASIA, American Spinal Cord Injury Association.

**Table 6 jcm-10-05664-t006:** Comparison of the conventional procedure and LAVR.

	Conventional Procedure	LAVR
Skin incision		
Thoracic/thoracolumbar	~20 cm (along the rib resection)	5 cm (rib partial resection)
Lumbar	~20 cm	5 cm (similar to XLIF approach)
Retractor	Conventional retractor	Dedicated retractor (MaXcess retractor)
Field of vision	Broad	Limited
Approach		
Thoracic/thoracolumbar	Thoracotomy Extrapleural	Extrapleural Transdiaphragm
Lumbar	Retroperitoneal	Retroperitoneal
Available cage	Cylindrical footplate cage	Rectangular footplate cage (X-Core2)
Operation time	>	
Blood loss	>	
2-year correction loss	>	
2-year bony fusion	<	
Learning curve		Steep

LAVR, lateral access vertebral reconstruction; XLIF, extreme lateral interbody fusion.

## Data Availability

The data presented in this study are available on request from the corresponding author. The data are not publicly available due to privacy or ethical restrictions.

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
