# Peer review of "Comparison of Outcomes between Minimally Invasive Lateral Approach Vertebral Reconstruction Using a Rectangular Footplate Cage and Conventional Procedure Using a Cylindrical Footplate Cage for Osteoporotic Vertebral Fracture"

_jcm, 2021, doi:10.3390/jcm10235664_

Round 1

Reviewer 1 Report

The given manuscript is well written. The topic is of interest to a wide part of medical doctors and to all spine surgeons. The content is new and of clinical relevance. However, some questions remain to material and methods and some facts should be further discussed. Please find my comments for detailed suggestions on improving the manuscript.

Material and Methods:

  • What cylindric footplate cages did you use?
  • Fig. 1: Do you mean: ≥65 years old
  • I think you used the Nuvasive MAXESS-retractor in Group L. Since the surgical approach depends much on this retractor, you should mention the name of the retractor.
  • Why did you perform a partial removal of the rib in Group L. This is not necessary. If you mobilize the rib, you can use the retractor without removing the rib. Please give a comment, why you are moving the rib.
  • Did you place a chest drain in patients of Group L in case of thoracotomy? Please give information on that.
  • How is the LLA defined? Is it always a bisegmental angle?

Results:

  • Readers may wonder about your operation time in Group L. Please compare your operation time to other groups using the same surgical approach (DOI: 10.1097/BRS.0000000000001862). Then, you can discuss that preparation for X-Core2-Cage needs more time, since both discs had to be removed almost completely.

Discussion:

  • You excluded 21 cases due to reduced follow up period. This may be a relevant selection bias of your study. Please give more information on these patients (e. g. how many of them are Group C or Group L) and discuss this selection bias in the discussion section.
  • You found a significant decrease in cage subsidence in Group L compared to Group C. This findings may be based on cage's different endplate sizes. Please include the current findings about the influence of cages' endplate sizes in osteoporotic bones in your discussion (https://doi.org/10.1016/j.clinbiomech.2020.105251)

Reviewer 2 Report

Well written paper.

I have some comments that I think would improve your paper.

Major

- One critical aspect that was not clear to me was indication. When was this operation performed, based on the German orthopedic association classification were all these fractures OF4 and OF5 types? 

- I would add a table that describes the differences between the 2 techniques.

- did you start osteoporotic treatment in patients that were diagnosed and not treated preoperatively. This could have an impact on subsidence rates

Minor

- in the abstract "In anterior surgery, 24 the operation time was significantly shorter (248 vs. 183 min, p < 0.001) and the blood loss was 25 significantly less (406 vs. 148 mL, p = 0.01) in Group L than in Group C." 

I would put reverse the numbers in parenthesis (ie 183 vs 248 min, p <0.001)

- Can you provide info on why hooks were used sometimes and not other times

- please include in the figure a preop and postop images of the 2 techniques
